# Coupled Fredkin and Motzkin chains from quantum six- and nineteen-vertex models

Zhao Zhang[1⋆] and Israel Klich[2]

**1** SISSA and INFN, Sezione di Trieste, via Bonomea 265, I-34136, Trieste, Italy
**2** Department of Physics, University of Virginia, Charlottesville, VA, USA

⋆ zhao.zhang@su.se

## Abstract

We generalize the area-law violating models of Fredkin and Motzkin spin chains into two dimensions by building quantum six- and nineteen-vertex models with correlated interactions. The Hamiltonian is frustration free, and its projectors generate ergodic dynamics within the subspace of height configuration that are non negative. The ground state is a volume- and color-weighted superposition of classical bi-color vertex configurations with non-negative heights in the bulk and zero height on the boundary. The entanglement entropy between subsystems has a phase transition as the $q$-deformation parameter is tuned, which is shown to be robust in the presence of an external field acting on the color degree of freedom. The ground state undergoes a quantum phase transition between area- and volume-law entanglement phases with a critical point where entanglement entropy scales as a function $L \log L$ of the linear system size $L$. Intermediate power law scalings between $L \log L$ and $L^2$ can be achieved with an inhomogeneous deformation parameter that approaches 1 at different rates in the thermodynamic limit. For the $q > 1$ phase, we construct a variational wave function that establishes an upper bound on the spectral gap that scales as $q^{-L^3/8}$.



# 1 Introduction

Entanglement entropy (EE) and its scaling has been a central theme of quantum many-body physics [1], not only because entanglement is a unique feature in the quantum world by itself, but also for their crucial role in determining the computational complexity of the numerical simulations of quantum many-body systems [2], indication of topological order [3] and understanding of the holographic principle and black hole entropy [4]. While EE of a generic eigenstate in the Hilbert space is shown to scale with the systems size [5], EE of the ground states of gapped local Hamiltonians are generally observed to obey the so-called area-law, scaling with the size of the boundary. A milestone in the study of area-law has been Hastings' rigorous proof of the result in one-dimensional systems [6]. Recently, a similar result in two-dimension has been proven for frustration-free models [7]. While area-law has been ubiquitous in gapped systems, plenty of examples of area-law violation has also been found in various gapless systems. (1+1)-dimensional critical system described by a conformal field theory has EE of logarithmic scaling [8,9]. EE of a system consisting of free fermions with a Fermi sea in dimension $d$ scales as $L^{d-1} \log L$ [10]. On the other hand, violations beyond logarithmic have only been known in one dimension so far.[1]

Quantum vertex and height models are an invaluable tool for the description of phases of strongly correlated systems [12–15]. They often emerge as an efficient description of quantum dimer models where strong local constraints facilitate the existence of a well defined "height" degree of freedom. In such models, a ground state may be well described in terms of a height field and its fluctuations. When coupled to other local degrees of freedom in such a way that the height field remains single valued, it is possible to enrich the model to use the fluctuating height field in order to further mediate correlations. One of the most spectacular examples of such a behavior is exhibited in the colored Motzkin and Fredkin spin chains [16–22], where the height degree of freedom can assist in generating an extensive entanglement entropy in the ground state. Such ground states thus exhibit a maximal violation of entanglement "area law". It is important to note that the degrees of freedom associated with the height field, due to continuity constraints, although playing a crucial role in facilitating the entanglement contribution from the color degree of freedom to be discussed in this work, cannot solely reproduce such area violation in higher dimensions by itself [23]. In this paper we construct a bicolor six- and nineteen-vertex models that admit exactly such behavior, in analogy with the recent lozenge tiling based model we have presented [24]. Our models are frustration free, with ground states being superpositions of surfaces with colorings, when viewed along a horizontal or vertical direction, obeying the coloring rules of arrays of colored Fredkin or Motzkin spin chains.

The paper is organized as follows. In Sec. 2, we quickly review the definition of Fredkin and Motzkin chains in one dimension and their common entanglement phase diagram. In Sec. 3, we first introduce the six-vertex construction of coupled Fredkin chains, with the Hamiltonian and its ground state explicitly written. In Sec. 4, the EE scaling of the ground state is extracted from a field theory description of the random surfaces in the ground state superposition, showing an entanglement phase transition of the $q$-deformation parameter. In Sec. 5, an upper bound on the spectral gap of the highly entangled phase is provided with a variational wave function. Sec. 6 sketches a similar nineteen-vertex construction of coupled Motzkin chains with similar EE scalings based on the previous sections. Finally, a summary and discussions of future direction are given in Sec. 7.

---

[1]A recent model with extensive entanglement has been found with a different mechanism in two dimensional space, but the lattice of the system has Hausdorff dimension one [11].

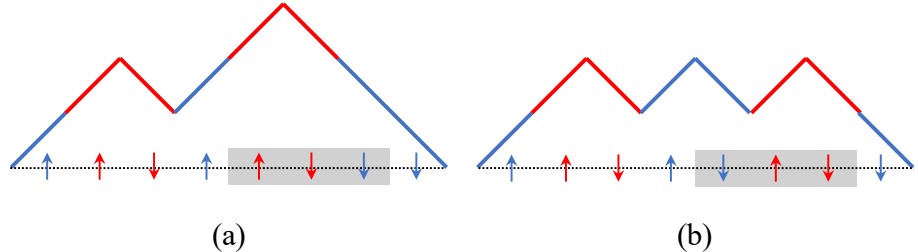

(a)                          (b)

Figure 1: Two configurations of a colored Fredkin chain of length 8 that differ by the two local configurations related by the projector $\left|F_{2,6}^{\mathrm{r,r,b}}\right\rangle\left\langle F_{2,6}^{\mathrm{r,r,b}}\right|$, which appear in the ground state superposition with a relative weight of $q$.

## 2 Review of the colored Fredkin and Motzkin chains

The colored Fredkin spin chain [20–22] has a local Hilbert space of spin-$\frac{1}{2}$ with two colors (red and blue) for up and down spins. It has a unique ground state as a superposition of colored Dyck walks/paths, which are random walks starting and ending at the origin and staying on one side in between, when spin up and down are mapped to an up and down move respectively, as depicted in Fig. 1. Its parent Hamiltonian consists of projection operators designed to make the ground state orthogonal to the projection onto the vectors

$$
\begin{aligned}
\left|F_{1,j}^{c_1,c_2,c_3}\right\rangle &= q^{-\frac{1}{2}}\left|\uparrow_{j-1}^{c_1}\uparrow_j^{c_2}\downarrow_{j+1}^{c_3}\right\rangle - q^{\frac{1}{2}}\left|\uparrow_j^{c_2}\downarrow_{j+1}^{c_3}\uparrow_{j+2}^{c_1}\right\rangle, \\
\left|F_{2,j}^{c_1,c_2,c_3}\right\rangle &= q^{-\frac{1}{2}}\left|\uparrow_{j-1}^{c_1}\downarrow_j^{c_2}\downarrow_{j+1}^{c_3}\right\rangle - q^{\frac{1}{2}}\left|\downarrow_{j-1}^{c_3}\uparrow_j^{c_1}\downarrow_{j+1}^{c_2}\right\rangle,
\end{aligned}
\tag{1}
$$

in the spin or height sector to enforce weighted superposition of Dyck paths of different height between the $j$'th and $(j+1)$'th spin, and

$$
\left|C_j\right\rangle = \left|\uparrow_j\downarrow_{j+1}\right\rangle - \left|\uparrow_j\downarrow_{j+1}\right\rangle,
\tag{2}
$$

in the color sector to enforce a balanced mixture of coloring of neighboring up-down pairs. Together with the projectors that applies energy penalty on color mismatching in the bulk and starting or ending the chain in the wrong direction, the Hamiltonian of the colored Fredkin chain is

$$
\begin{aligned}
H_{\mathrm{F}} = &\sum_{j=2}^{L-1}\sum_{a=1}^{2}\sum_{c_1,c_2,c_3=\mathrm{r,b}}\frac{1}{[2]_q}\left|F_{a,j}^{c_1,c_2,c_3}\right\rangle\left\langle F_{a,j}^{c_1,c_2,c_3}\right| \\
&+\sum_{j=1}^{L-1}\left(\frac{1}{2}\left|C_j\right\rangle\left\langle C_j\right| + \left|\uparrow_j\downarrow_{j+1}\right\rangle\left\langle\uparrow_j\downarrow_{j+1}\right| + \left|\uparrow_j\downarrow_{j+1}\right\rangle\left\langle\uparrow_j\downarrow_{j+1}\right|\right) \\
&+\sum_{c=\mathrm{r,b}}\left(\left|\downarrow_1^c\right\rangle\left\langle\downarrow_1^c\right| + \left|\uparrow_L^c\right\rangle\left\langle\uparrow_L^c\right|\right),
\end{aligned}
\tag{3}
$$

where the q-deformed integer 2 is defined as $[2]_q := q + q^{-1}$. Its ground state can be written as

$$
\left|\mathrm{GS}_{\mathrm{F}}\right\rangle = \frac{1}{\mathcal{N}_{\mathrm{F}}}\sum_{w\in\text{colored Dyck walks}}q^{\frac{1}{2}\mathcal{A}(w)}\left|w\right\rangle,
\tag{4}
$$

where $\mathcal{N}_{\mathrm{F}}$ is the normalization constant and $\mathcal{A}(w)$ denotes the area underneath the Dyck path $w$.

The integer spin counterpart of colored Fredkin chain is called colored Motzkin chain [17, 18], for its ground state is a superposition of Motzkin paths/walks, which are Dyck paths

$$S_{\frac{L}{2}} \sim \mathcal{O}(1) \qquad S_{\frac{L}{2}} \sim \sqrt{L} \qquad S_{\frac{L}{2}} \sim L \log 2$$

$$0 \qquad\qquad\qquad 1 \qquad\qquad\qquad q$$

Figure 2: The common entanglement phase diagram of colored Fredkin and Motzkin chains.

diluted with spin-0's or flat steps. The bulk Hamiltonian projects onto the vectors orthogonal to the superposition in the ground state

$$\begin{aligned}
\left| M_{1,j}^{c} \right\rangle &= q^{-\frac{1}{2}} \left| \uparrow_{j}^{c} \, 0_{j+1} \right\rangle - q^{\frac{1}{2}} \left| 0_j \, \uparrow_{j+1}^{c} \right\rangle, \\
\left| M_{2,j}^{c} \right\rangle &= q^{-\frac{1}{2}} \left| 0_j \, \downarrow_{j+1}^{c} \right\rangle - q^{\frac{1}{2}} \left| \downarrow_{j}^{c} \, 0_{j+1} \right\rangle, \\
\left| M_{3,j}^{c} \right\rangle &= q^{-\frac{1}{2}} \left| \uparrow_{j}^{c} \downarrow_{j+1}^{c} \right\rangle - q^{\frac{1}{2}} \left| 0_j 0_{j+1} \right\rangle.
\end{aligned} \tag{5}$$

As color matching is automatically enforced, the colored Motzkin Hamiltonian is just the sum over these bulk terms and the same boundary terms as the Fredkin Hamiltonian.

$$H_{\mathrm{M}} = \sum_{c=\mathrm{r,b}} \left( \sum_{j=2}^{L-1} \sum_{a=1}^{3} \frac{1}{[2]_q} \left| M_{a,j}^{c} \right\rangle \left\langle M_{a,j}^{c} \right| + \left| \downarrow_{1}^{c} \right\rangle \left\langle \downarrow_{1}^{c} \right| + \left| \uparrow_{L}^{c} \right\rangle \left\langle \uparrow_{L}^{c} \right| \right). \tag{6}$$

The ground state of the colored Motzkin chain is given as

$$\left| \mathrm{GS_M} \right\rangle = \frac{1}{\mathcal{N}_{\mathrm{M}}} \sum_{w \in \text{colored Motzkin walks}} q^{\mathcal{A}(w)} |w\rangle, \tag{7}$$

where $\mathcal{N}_{\mathrm{M}}$ is the normalization constant and $\mathcal{A}(w)$ denotes the area underneath the Motzkin path $w$.

When a cut in the middle separates the chain into two subsystems, both of the ground states (4) and (7) can be Schmidt decomposed by the height $m$ of the path in the middle at the cut, and the coloring of the $m$ spins in one of the subsystems, which is to be matched with those in the other subsystem

$$\left| \mathrm{GS_{F(M)}} \right\rangle = \sum_{m=0}^{L/2} \sqrt{p_m} \sum_{c_1,\ldots,c_m=\mathrm{r,b}} \left| \mathcal{W}_m^{\vec{c}} \right\rangle_{\mathrm{L}} \otimes \left| \mathcal{W}_m^{\vec{c}} \right\rangle_{\mathrm{R}}, \tag{8}$$

where $\left| \mathcal{W}_m^{\vec{c}} \right\rangle_{\mathrm{L,R}}$ are the normalized superposition of all paths on the left and right subsystems that have height $m$ in the middle point. Any such walk will have $m$ uncompensated up spins on the left whose colors are exactly matched with $m$ down spins respectively on the right subsystem. The vector $\vec{c}$ specifies the colors of the extra up steps on the left. The Schmidt coefficient $p_m$ determines the EE of the half chain

$$S_{\frac{L}{2}} = -\sum_{m=0}^{L/2} 2^m p_m \log p_m. \tag{9}$$

Detailed analysis of the probability distribution of $m$ or its weighted average among the Motzkin and Fredkin walks shows that the two model share the same phase diagram 2, characterized by the scaling of half chain EE [17, 18, 20–22].

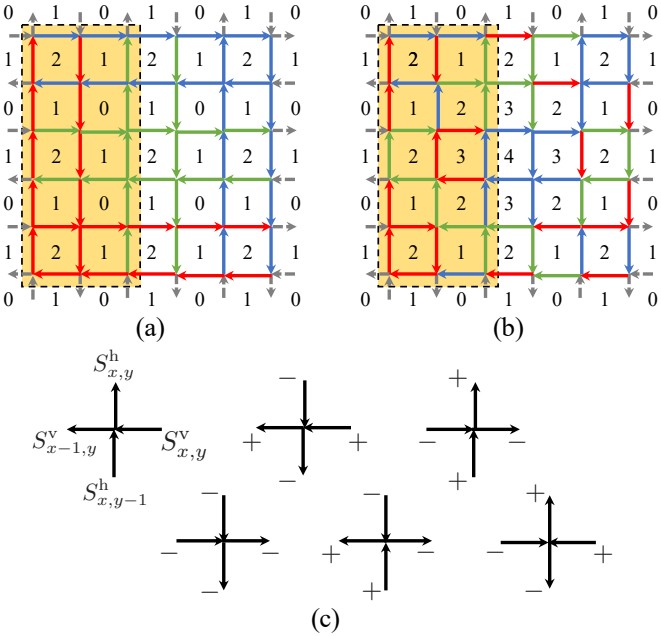

Figure 3: (a) A coloring of the minimal height configuration configuration of a lattice of linear size 6. Three colors are used to manifest the matching pattern. The shaded area with dashed line boundary marks the subsystem A, where the location of the cut giving rise to the Schmidt decomposition (19) is along the left coloumn of vertical spins. Then numbers in the plaquettes indicate the height configuration in the dual lattice. (In both (a) and (b), an additional green color is used to illustrate the pairing of up-down spins at the same hight that match in color, even though in the rest of the paper, the local Hilbert space of the model is defined with only red and blue color.) (b) A coloring of the maximal height configuration. The heights in the i'th square of plaquettes counting from boundary inward alternates between $i$ and $i + 1$.(c) The convention of positive direction of the spins living on the array of horizontal chains $S^{\mathrm{h}}$ and vertical chains $S^{\mathrm{v}}$ at vertex $(i, j)$, along with the other 5 allowed vertex configurations.

## 3 Six-vertex construction of coupled Fredkin chains

The degrees of freedom of the model live on the edges or bonds between vertices of a square lattice of linear size $L$. They can be decomposed into two arrays of one-dimensional spin-$\frac{1}{2}$ chains, one horizontally and one vertically aligned. Each of the edges in the array of horizontal (resp. vertical) chains can have a spin $S^{\mathrm{h}}$ (resp. $S^{\mathrm{v}}$) either up or down (resp. left or right) corresponding to $\pm \frac{1}{2}$. These two sets of degrees of freedom are coupled to each other by the ice rule [25, 26] in Fig. 3 (c), enforced by the bulk local Hamiltonian

$$H_0 = \sum_{x,y=1}^{L} \left( S^{\mathrm{h}}_{x,y-1} - S^{\mathrm{h}}_{x,y} - S^{\mathrm{v}}_{x-1,y} + S^{\mathrm{v}}_{x,y} \right)^2 . \tag{10}$$

The spins $S^{\mathrm{v}}_{0,y}$ and $S^{\mathrm{v}}_{L,y}$, $y = 1, \cdots, L$ and $S^{\mathrm{h}}_{x,0}$ and $S^{\mathrm{h}}_{x,L}$, $x = 1, \cdots, L$ are auxiliary spins fixed to be alternating ups and downs as shown in Fig. 3 (a) and (b), and are not part of the degrees of freedom of the system. The global Hilbert space can be constrained to the subspace of six-vertex configurations by making the coefficient of this term $V_0 \gg 1$. The boundary spins can

be fixed by the Hamiltonian

$$H_\partial = \sum_{y=1}^{L-1}\left(S_{L,y}^{\mathrm{h}} - S_{1,y}^{\mathrm{h}}\right) + \sum_{x=1}^{L-1}\left(S_{x,L}^{\mathrm{v}} - S_{x,1}^{\mathrm{v}}\right), \tag{11}$$

such that the boundary configurations in Fig. 3 has the minimal energy of 0, and any other configurations will be penalized in proportion to the number of local differences from them along the boundary.

The six-vertex condition allows a well-defined height function $\phi_{x+\frac{1}{2},y+\frac{1}{2}}$ living on the dual lattice of plaquette centers satisfying the rules according to the convention in Fig. 3 (c)

$$\phi_{x+\frac{1}{2},y+\frac{1}{2}} - \phi_{x+\frac{1}{2},y-\frac{1}{2}} = 2S_{x,y}^{\mathrm{v}}, \tag{12}$$

$$\phi_{x+\frac{1}{2},y+\frac{1}{2}} - \phi_{x-\frac{1}{2},y+\frac{1}{2}} = 2S_{x,y}^{\mathrm{h}}, \tag{13}$$

up to a global gauge transformation of shifting the heights by a constant. For convenience we will fix the gauge so that the height $\phi_{0,0} = 0$ at the lower left corner of the lattice. The effect of boundary Hamiltonian amounts to picking a (Dirichlet) boundary condition on the height for the ground state wave-function, as shown in Fig. 3(a), Fig. 3(b) where the height function alternates between 0 and 1 along the boundary.

To enrich the entanglement of the ground state, the local Hilbert space of each spin is further enlarged to have either a red or blue color, along with a local Hamiltonian $H_C$ between neighboring up-down spin pairs to match in color

$$
\begin{aligned}
H_C = \sum_{x,y=1}^{L-1}\Bigg( & |{\uparrow_x\downarrow_{x+1}}\rangle_y \langle{\uparrow_x\downarrow_{x+1}}| + |{\uparrow_x\downarrow_{x+1}}\rangle_y \langle{\uparrow_x\downarrow_{x+1}}| + \left|{\overset{\to y+1}{\leftarrow y}}\right\rangle_x \left\langle{\overset{\to y+1}{\leftarrow y}}\right| + \left|{\overset{\to y+1}{\leftarrow y}}\right\rangle_x \left\langle{\overset{\to y+1}{\leftarrow y}}\right| \\
& + \frac{1}{[2]_r}\left(r^{-\frac{1}{2}}|{\uparrow_x\downarrow_{x+1}}\rangle_y - r^{\frac{1}{2}}|{\uparrow_x\downarrow_{x+1}}\rangle_y\right)\left(r^{-\frac{1}{2}}\langle{\uparrow_x\downarrow_{x+1}}|_y - r^{\frac{1}{2}}\langle{\uparrow_x\downarrow_{x+1}}|_y\right) \\
& + \frac{1}{[2]_r}\left(r^{-\frac{1}{2}}\left|{\overset{\to y+1}{\leftarrow y}}\right\rangle_x - r^{\frac{1}{2}}\left|{\overset{\to y+1}{\leftarrow y}}\right\rangle_x\right)\left(r^{-\frac{1}{2}}\left\langle{\overset{\to y+1}{\leftarrow y}}\right|_x - r^{\frac{1}{2}}\left\langle{\overset{\to y+1}{\leftarrow y}}\right|_x\right)\Bigg),
\end{aligned}
\tag{14}
$$

where the up and down (resp. left and right) arrows are used to denote spin $\frac{1}{2}$ in the horizontal (resp. vertical) direction. The terms in the first line energetically penalize adjacent up-down spin pairs mismatching in color, so that spin configurations containing, say $|{\uparrow_x\downarrow_{x+1}}\rangle_y^{\mathrm{h}}$ do not appear in the spin configuration of the zero energy ground state. Two colored spin configurations that are not penalized by the mismatch penalty terms in $H_C$ are examplified in Fig. 3 (a) and (b), where an additional color green is employed to better illustrate the color matching between up-down pairs of the same height. The terms in the next two lines enforce a superposition of colorings of such adjacent up-down spin pairs when their color is matched, tuned by the deformation parameter $0 < r < 1$. Indeed, whenever a spin/color configuration $|{\uparrow_x\downarrow_{x+1}}\rangle_y$ appears in the ground state, it must appear through the combination $r^{\frac{1}{2}}|{\uparrow_x\downarrow_{x+1}}\rangle_y + r^{-\frac{1}{2}}|{\uparrow_x\downarrow_{x+1}}\rangle_y$ in order to a avoid an energy penalty from these projection operators. In this way, these projection operators are necessary to provide color mixing and ergodicity within the subspace of product states annihilated by the terms in the first line.

The deformation parameter $r$ plays the role of an external color field, such that when $r = 1$, the ground state will have a uniform superposition of different coloring, while when $r > 1$, the configurations with more red colored spins will be favored.

Since the color Hamiltonian only acts on up-down and left-right pairs, for it to affect all the spins in the system, there must be a net surplus of up (resp. left) spins in any sub-chain counting from left (resp. bottom). In other words, the height function in the dual lattice must

stay non-negative and the spins form Dyck paths along the chains in both directions. This can be enforced by the correlated swapping Hamiltonian

$$H_S = \sum_{x,y=2}^{L-1} \sum_{f_{\mathrm{h}},f_{\mathrm{v}}=\pm} \sum_{c_1,...,c_6=\mathrm{r,b}} \frac{1}{[2]_q} \left| \pi_{x,y,f_{\mathrm{h}},f_{\mathrm{v}}}^{c_1,...,c_6} \right\rangle \left\langle \pi_{x,y,f_{\mathrm{h}},f_{\mathrm{v}}}^{c_1,...,c_6} \right|,$$

(15)

where

$$
\begin{aligned}
\left| \pi_{x,y,-,+}^{c_1,...,c_6} \right\rangle &= q^{-\frac{1}{2}} \left| \begin{array}{c} \end{array} \right\rangle - q^{\frac{1}{2}} \left| \begin{array}{c} \end{array} \right\rangle, \\
\left| \pi_{x,y,+,+}^{c_1,...,c_6} \right\rangle &= q^{-\frac{1}{2}} \left| \begin{array}{c} \end{array} \right\rangle - q^{\frac{1}{2}} \left| \begin{array}{c} \end{array} \right\rangle, \\
\left| \pi_{x,y,-,-}^{c_1,...,c_6} \right\rangle &= q^{-\frac{1}{2}} \left| \begin{array}{c} \end{array} \right\rangle - q^{\frac{1}{2}} \left| \begin{array}{c} \end{array} \right\rangle, \\
\left| \pi_{x,y,+,-}^{c_1,...,c_6} \right\rangle &= q^{-\frac{1}{2}} \left| \begin{array}{c} \end{array} \right\rangle - q^{\frac{1}{2}} \left| \begin{array}{c} \end{array} \right\rangle.
\end{aligned}
$$

(16)

The total Hamiltonian

$$H_{cF} = H_0 + H_\partial + H_C + H_S,$$

(17)

is a frustration-free sum of projection operators, meaning its zero energy ground state is the simultaneous lowest energy eigenstate of each term. Since each term in the Hamiltonian requires a superposition of locally different height and coloring in a particular way, the ground state is therefore a weighted superposition of bicolored six-vertex configurations with alternating heights between 0 and 1 along the boundary, and non-negative heights in the bulk

$$|\mathrm{GS}\rangle = \frac{1}{\sqrt{\mathcal{N}}} \sum_{\phi(\partial P)=\frac{1}{2}} \prod_{x,y=1}^{L} \theta(\phi_{x,y}) \sideset{}{'}\sum_{C} r^{\frac{M(C)}{2}} q^{\frac{\mathcal{V}(P)}{2}} \left| P^C \right\rangle,$$

(18)

where for simplified notation, the height function of each spin is taken to be the average between the heights of its two adjacent plaquettes, the first sum is over all six-vertex configurations $P$ with boundary height $\frac{1}{2}$, the second primed sum is over coloring patterns with spins in the same chain on the same height matching. $\theta$ is the Heaviside step function, indicating the sum is over configurations with non-negative height in the bulk. The volume of a configuration is defined as $\mathcal{V}(P) = \sum_{x,y=1}^{L-1} \phi_{x+\frac{1}{2},y+\frac{1}{2}}$, $M(C)$ is the "warmness" magnetization of coloring $C$, defined as the difference between the number of pairs of red and blue spins, and $\mathcal{N}$ is the normalization constant that depends only on $q$ and $r$. The uniqueness of the ground state is guaranteed by the ergodicity of the Hamiltonian (15), which is proven in the appendix.

## 4  Scaling of entanglement entropy

The model has an apparent $D_4$ lattice symmetry, so a cut across the middle along either the horizontal or vertical direction gives the same bipartite entanglement entropy between subsystems. Unlike a quasi-2D model of trivial stacking an array of Fredkin or Motzkin chains,

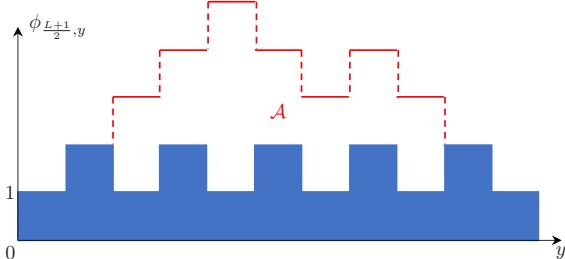

Figure 4: Cross-sectional view of the stepped surface outlined by the height function along the middle cut. Area $\mathcal{A}$ is defined as the sum of heights at each step counting from the minimal heights in Fig. 3 (a).

the ground state EE scaling behavior of this coupled 2D model is the same for cuts in any direction. Without loss of generality, we choose a vertical cut as shown in Fig. 3. Just like the one-dimensional model described in Sec. 2, the entanglement comes from the extra up spins in the left subsystem, or equivalently the surplus of down spins in the right subsystem. Here, when performing the Schmidt decomposition on the ground state (18), we need to keep track of the heights of each chain in the array along the cut in the middle, denoted by the vector $\vec{\phi}_{\frac{L+1}{2}}$, as well as the vectors $\vec{c}_y$'s denoting the colors of the spins in each row to be matched between the two subsystem. The number of components of each vector $\vec{c}_y$ is given by the value of the component of height vector $\phi_{\frac{L+1}{2},y+\frac{1}{2}}$. Notice that $\vec{\phi}_{\frac{L+1}{2}}$ itself is constrained to be a Dyck path in the vertical direction, the bulk components of which are always larger than or equal to the first and last components $\phi_{\frac{L+1}{2},\frac{1}{2}} \equiv \phi_{\frac{L+1}{2},L+\frac{1}{2}} = 0, 1$. Therefore the Schmidt decompostion can be written as

$$|GS\rangle = \sum_{\vec{\phi}_{\frac{L+1}{2}} \in \text{Dyck paths}} \sqrt{\frac{\mathcal{M}_{\vec{\phi}}^2}{\mathcal{N}}} \sum_{\vec{c}_1 \in \{r,b\}^{\phi_{\frac{L+1}{2},3/2}}} \cdots \sum_{\vec{c}_L \in \{r,b\}^{\phi_{\frac{L+1}{2},L+1/2}}} \left( \prod_{y=1}^{L} r^{\frac{m(\vec{c}_y)}{2}} \right) \left| P_{\vec{\phi}_{\frac{L+1}{2}}}^{\vec{c}_1,...,\vec{c}_L} \right\rangle_L \otimes \left| P_{\vec{\phi}_{\frac{L+1}{2}}}^{\vec{c}_1,...,\vec{c}_L} \right\rangle_R , \quad (19)$$

where the "warmness" magnetization $m(\vec{c}_y)$ of the unmatched colors of the half system in the y'th chain is the difference between the number of red and blue spin pairs among them, and

$$\left| P_{\vec{\phi}_{\frac{L+1}{2}}}^{\vec{c}_1,...,\vec{c}_L} \right\rangle_{L(R)} = \frac{1}{\sqrt{\mathcal{M}_{\vec{\phi}_{\frac{L+1}{2}}}}} \sum{}' r^{\frac{m_{L(R)}}{2}} q^{\frac{v_{L(R)}(P^C)}{2}} \left| P^C \right\rangle_{L(R)} \quad (20)$$

are normalized wave functions of the left (resp. right) subsystems, the primed sum is a shorthand notation for summing over six-vertex configurations with non-negative height in the bulk plaquettes and in particular of height specified by $\vec{\phi}_{\frac{L+1}{2}}$ on the middle boundary. $m_{L(R)}$ is the redness magnetization of the pairs with color matched within the subsystem, which takes value between 0 and $(L^2 - L - \frac{A(\vec{\phi}_{\frac{L+1}{2}})+3L}{2} + 1)/2$, with $A(\vec{\phi}_{\frac{L+1}{2}}) = \sum_{y=1}^{L-1} \phi_{\frac{L+1}{2},y+1/2}$ being the cross-sectional area of the stepped surface outlined by the height function, as depicted in Fig. 4.

The normalization constants are given by

$$\mathcal{M}^2_{\vec{\phi}_{\frac{L+1}{2}}} = [2]_r^{L^2 - \frac{5L}{2} - \frac{\mathcal{A}(\vec{\phi}_{\frac{L+1}{2}})}{2} + 1} \sum_{\vec{\phi}_{\frac{L+1}{2}}(T|_{\text{cut}}) = \vec{\phi}_{\frac{L+1}{2}}} q^{\mathcal{V}(P)}, \tag{21}$$

and

$$\mathcal{N} = \sum_{\vec{\phi}} [2]_r^{\frac{\mathcal{A}(\vec{\phi}_{\frac{L+1}{2}}) + 3L}{2} - 1} \mathcal{M}^2_{\vec{\phi}_{\frac{L+1}{2}}} \tag{22}$$

$$\equiv [2]_r^{L^2 - L} \sum_{\phi(\partial P) = \frac{1}{2}} q^{\mathcal{V}(P)}. \tag{23}$$

The Schmidt coefficients are given by the probability of height configuration $\vec{\phi}$ with coloring $\{\vec{c}_1, ..., \vec{c}_L\}$ of the cross-section between subsystems

$$p\left(\vec{\phi}_{\frac{L+1}{2}}, \{\vec{c}_1, ..., \vec{c}_L\}\right) = \frac{\left(\prod_{y=1}^{L} r^{\frac{m(\vec{c}_y)}{2}}\right)^2 \mathcal{M}^2_{\vec{\phi}_{\frac{L+1}{2}}}}{\mathcal{N}} = p\left(\{\vec{c}_1, ..., \vec{c}_L\} \mid \vec{\phi}_{\frac{L+1}{2}}\right) p\left(\vec{\phi}_{\frac{L+1}{2}}\right), \tag{24}$$

with

$$p\left(\{\vec{c}_1, ..., \vec{c}_L\} \mid \vec{\phi}_{\frac{L+1}{2}}\right) = [2]_r^{-\frac{\mathcal{A}(\vec{\phi}_{\frac{L+1}{2}})}{2} - \frac{3L}{2} + 1} \prod_{y=1}^{L} r^{m(\vec{c}_y)}, \tag{25}$$

can be factorized as a product of probability of having a particular coloring $\{\vec{c}_1, ..., \vec{c}_L\}$ of the unmatched pairs within the subsystems, conditioned on having a Dyck path $\vec{\phi}_{\frac{L+1}{2}}$ along the cut, and the marginal probability $p(\vec{\phi}_{\frac{L+1}{2}}) \equiv \sum_{\vec{c}_1} \cdots \sum_{\vec{c}_L} p(\phi_{\frac{L+1}{2}}, \{\vec{c}_1, ..., \vec{c}_L\})$ of finding such a cross section among uncolored random height configurations. The entanglement entropy decomposed into a piece given in terms of average cross-sectional area of a random height configuration, and another subleading contribution from the fluctuation of the random surface

$$\begin{aligned} S_{\frac{L}{2} \times L}(q, r) &= -\sum_{\vec{\phi}_{\frac{L+1}{2}}} \sum_{\vec{c}_1} \cdots \sum_{\vec{c}_L} p\left(\vec{\phi}_{\frac{L+1}{2}}, \{\vec{c}_1, ..., \vec{c}_L\}\right) \log p\left(\vec{\phi}, \{\vec{c}_1, ..., \vec{c}_L\}\right) \\ &= \sum_{\vec{\phi}_{\frac{L+1}{2}}} p\left(\vec{\phi}_{\frac{L+1}{2}}\right) S^c_{\frac{L}{2} \times L}(\vec{\phi}_{\frac{L+1}{2}}, r) + S^{\phi}_{\frac{L}{2} \times L}(q) \\ &= \frac{C_r}{2}(\langle \mathcal{A} \rangle + 3L - 2) + S^{\phi}_L(q), \end{aligned} \tag{26}$$

where $S^{\phi}_{\frac{L}{2} \times L}(q) = -\sum_{\vec{\phi}_{\frac{L+1}{2}}} p(\vec{\phi}_{\frac{L+1}{2}}) \log p(\vec{\phi}_{\frac{L+1}{2}})$ and in third line we have used

$$\begin{aligned} S^c_{\frac{L}{2} \times L}(\vec{\phi}_{\frac{L+1}{2}}, r) &= -\sum_{\{\vec{c}_1, ..., \vec{c}_L\}} p\left(\{\vec{c}_1, ..., \vec{c}_L\} \mid \vec{\phi}_{\frac{L+1}{2}}\right) \log p\left(\{\vec{c}_1, ..., \vec{c}_L\} \mid \vec{\phi}_{\frac{L+1}{2}}\right) \\ &= -(\log r)[2]_r^{-\frac{\mathcal{A}(\vec{\phi}_{\frac{L+1}{2}})}{2} - \frac{3L}{2} + 1} \sum_{\{\vec{c}_1, ..., \vec{c}_L\}} \left(\sum_{y=1}^{L} m(\vec{c}_y)\right) \prod_{y=1}^{L} r^{m(\vec{c}_y)} \\ &\quad + \log[2]_r \left(\frac{\mathcal{A}(\vec{\phi}_{\frac{L+1}{2}})}{2} + \frac{3L}{2} - 1\right) \\ &= C_r\left(\frac{\mathcal{A}(\vec{\phi}_{\frac{L+1}{2}})}{2} + \frac{3L}{2} - 1\right). \end{aligned} \tag{27}$$

The sum in the second line can also be written as a sum over the total number of pairs of red spin among $\{\vec{c}_1, ..., \vec{c}_L\}$, $i \equiv \sum_{y=1}^{L} \frac{m(\vec{c}_y) + \phi_{\frac{L+1}{2}, y+\frac{1}{2}}}{2}$ which ranges from 0 to $N \equiv \frac{\mathcal{A}(\vec{\phi}_{\frac{L+1}{2}})}{2} + \frac{3L}{2} - 1$:

$$\sum_{\{\vec{c}_1,...,\vec{c}_L\}} \left( \sum_{y=1}^{L} m(\vec{c}_y) \right) \prod_{y=1}^{L} r^{m(\vec{c}_y)} = \sum_{i=0}^{N} \binom{N}{i}(2i-N)r^{2i-N}$$

$$= \frac{r - r^{-1}}{[2]_r}[2]_r^N N,$$

(28)

which gives the coefficient

$$C_r = -\frac{r - r^{-1}}{[2]_r} \log r + \log[2]_r.$$

(29)

This kind of decomposition of entanglement entropy as a result of enlarging the local Hilbert space has also been observed recently in the Bethe Ansatz integrable excited states of a non-integrable one-dimensional multicomponent spin chain [27], which emerges from certain phases of a quasi-2D spin ladder [28].

For any finite $r$, $C_r$ is a finite constant independent of $L$, so the problem is reduced to finding the scaling of the average area $\langle \mathcal{A} \rangle$. That can be done in a field theoretic fashion, as was previously used to study the dynamics of the one-dimensional Motzkin and Fredkin chains [29, 30]. A continuous field of the height configuration can be defined as a piece-wise linear function $\phi(x, y)$, which takes the value of $\phi_{x+1/2, y+1/2}$ on the dual lattice. It is well known that the "entropy" of random surface is captured by a surface tension $\sigma(\nabla\phi(x, y))$ as a function of the height gradient alone [31–34]. Also taking into account the "energy" contribution from volume weighting, we get the partition function

$$Z = \int \mathcal{D}\phi(x, y) e^{\iint dx dy (-\sigma(\nabla\phi(x,y)) + (\log q)\phi(x,y))},$$

(30)

where $\mathcal{D}\phi(x, y)$ is a continuous version of

$$\prod_{x,y} \int_0^{+\infty} d\phi_{x+1/2, y+1/2} \equiv \prod_v \int_{-\infty}^{+\infty} dh_v \theta(h_v),$$

(31)

and where $\phi$ obeys a Lipschitz condition $|\partial_{x'}\phi, \partial_{y'}\phi| \leq 1$, and $\theta$ is the Heaviside step function.

The linear contribution in $\phi$ is dominant when $q > 1$. To see this explicitly, we substitute

$$x = Lx', \quad y = Ly',$$

(32)

which makes

$$\nabla = L^{-1}\nabla', \quad dx = Ldx', \quad dy = Ldy'.$$

(33)

The free energy associated to a height configuration then becomes

$$F[\phi] = L^2 \iint_0^1 dx' dy' \left( \sigma\left( \frac{\nabla'\phi(x', y')}{L} \right) - (\log q)\phi(x', y') \right),$$

(34)

where $\phi(x', y') = \phi(x, y)$ now satisfy the Lipschitz property of $|\partial_{x'}\phi, \partial_{y'}\phi| \leq L$ instead. The surface tension term counts the entropy of height configurations associated with height variations in a small region with height differences $\partial_{x'}\phi, \partial_{y'}\phi$ on the boundary of the region, and is thus trivially bounded by the entropy density of ice. Thus, in the thermodynamic limit, the surface tension term becomes irrelevant compared to the linear term $(\log q)\phi(x', y')$ when $q \neq 1$. Therefore $F[\phi]$ is minimized by the Lipschitz property for the $q > 1$ case, where minimization

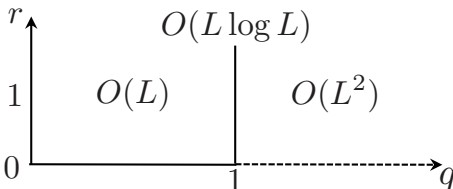

Figure 5: The entanglement phase diagram with two distinct EE scaling separated by the critical line at $q = 1$.

of $F$ is achieved with maximal gradient and maximal volume; and by the positivity for the $q < 1$ case, where $F$ is minimized taking $\phi(x, y) = 0$. Therefore, for $q$ larger and smaller than 1 respectively, we have $\langle \mathcal{A} \rangle = O(L^2)$ and $O(L)$. (26) then says $S_L(q, r)$ goes through a phase transition at $q = 1$ from volume scaling to area law.

At the critical point, the height field becomes a massless field conditioned on staying positive. Given that the surface tension is a strictly convex even function of the height variable [31–33, 35–38], the average height was rigorously shown to have the scaling $O(\log L)$, as a result of being repelled by the hard-wall at zero height [39]. This gives the same EE scaling of $O(L \log L)$ as in the recent quantum lozenge tiling model [40], despite the height field of uniform weighted six-vertex model being interacting and not described by a Gaussian free field. This entanglement phase transition can be summarized in the phase diagram in Fig. 5.

The stark phase transition for any $\epsilon = q - 1$ is a consequence of the discontinuity of the partition function $Z$ when the thermodynamic/scaling limit is taken. To obtain an intermediate scaling between $L \log L$ and $L^2$, one can consider a varying $q = e^{\lambda L^{-\alpha}}$ that approaches 1 as $L \to \infty$. Such scaling limits are of interest random surface models, as they admit existence of non-trivial limit shapes [41]. For $\alpha \in (1, 2)$, simple scaling argument gives an EE scaling of $L^{3-\alpha}$ with a $\lambda$ dependent coefficient. Whereas for $\alpha \geq 2$, it gives the $L \log L$ scaling, and for $\alpha \leq 1$, it gives the $L^2$ scaling. Interestingly, one can think of this intermediate entropy scaling as the scaling of entropy associated with a square neighbourhood of size $L'$ attached to the corner of a larger lattice where the deformation parameter is inhomogenous, decaying as a function $e^{\lambda d^{-\alpha}}$ of the distance $d$ from the corner of the lattice to the center.

# 5 Scaling of spectral gap

Following the strategy in the proof of the gaplessness of the highly entangled phase of the one-dimensional models [22, 42], we construct a variational wave function that has both a small overlap with the ground state (18) and an exponentially small expectation value of the Hamiltonian (17). Hence it inevitably implies that the spectral gap of the $q > 1$ phase is exponentially small and hence gapless in the thermodynamic limit.

We start by defining a subset $\mathcal{E}$ of the six-vertex configurations with non-negative height in the bulk, which will appear in the superposition of the trial excited state.

*Definition of $\mathcal{E}$.* A non-negative six-vertex configuration obeying the alternating Dirichlet condition of Fig. 3 belongs to $\mathcal{E}$ if: (i) the lowest height in the bulk of a configuration is either 0 or 1; and (ii), the *longest* distance between the lowest height in the bulk (be it 0 or 1) and any of the four sides of the boundary is larger than $\frac{L}{2}$.

An example of a configuration in this set, incidentally also one of the four such ones with lowest total volume, is shown in Fig. 6 (a). Whereas the configuration with largest volume among those not belonging to this set is given in Fig. 6 (b). Note that in this section, for clarity of presentation we have replaced the step-wise structure of the six-vertex height model with a linear interpolating representation that would more easily resemble the 1D construction [22, 42].

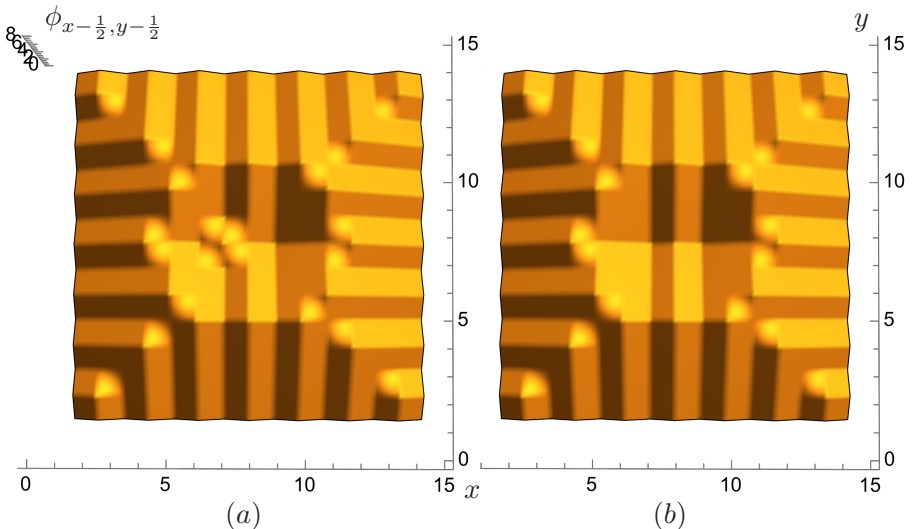

Figure 6: Two height configurations of a $14 \times 14$ system that differ by the height of one plaquette $(6 + \frac{1}{2}, 7 + \frac{1}{2})$, with the corresponding height profiles of the cross-section along the dashed line $(x, 7 + \frac{1}{2})$. (a) belongs to the set $\mathcal{E}$: the longest span of non-negative height is larger than $\frac{L}{2}$, while (b) doesn't.

The trial excited state is defined by changing the color of the spin on the endpoint inside the bulk along the said longest distance to the boundary from the plaquette with lowest height

$$|\text{ex}\rangle = \frac{1}{\sqrt{\mathcal{N}'}} \sum_{P \in \mathcal{E}} \sum_{C}'' r^{\frac{M(C)}{2}} q^{\frac{V(P)}{2}} \left| P^C \right\rangle, \tag{35}$$

where $\sqrt{\mathcal{N}'}$ is the new normalization constant, and compared to (18), the double primed sum over colored refers the matching of all the other pairs of spins at the same height in color except the one at the endpoint of the longest nonzero height streak that is now forced to mismatch in color, as examplified in Fig. 7 (b). Due to this mismatching, the excited state must be orthogonal to the ground state

$$\langle \text{ex} | \text{GS} \rangle = 0. \tag{36}$$

Furthermore, since all the configurations in $\mathcal{E}$ are of larger volume than the one with the longest streak containing the color mismatch, the two mismatched spins never appear as neighbors in the superposition, we have

$$H_0 |\text{ex}\rangle = 0, \quad H_\partial |\text{ex}\rangle = 0, \quad \text{and} \quad H_C |\text{ex}\rangle = 0. \tag{37}$$

So the non-vanishing contribution to the energy expectation can only come from $H_S$, precisely from the term acting on the plaquette, decreasing the height on which would result in a configuration outside the set $\mathcal{E}$. In the case of the configuration $P_a$ in Fig. 6 (a), the terms involved in $H_S$ will be

$$h_{6,7} = \sum_{c_1,\dots,c_6 = \text{r,b}} \frac{1}{[2]_q} \left( \left| \pi_{6,7,+,+}^{c_1,\dots,c_6} \right\rangle \left\langle \pi_{6,7,+,+}^{c_1,\dots,c_6} \right| + \left| \pi_{6,7,+,-}^{c_1,\dots,c_6} \right\rangle \left\langle \pi_{6,7,+,-}^{c_1,\dots,c_6} \right| \right). \tag{38}$$

Together they contribute $\langle P_a | h_{6,7} | P_a \rangle = \frac{2}{1+q^2}$, for each particular color configuration. The number of such height configurations that can be brought out of set $\mathcal{E}$ can be very roughly upper bounded by the total number of spin and color configurations $(4 \max\{(1 + r^2), (1 + r^{-2})\})^{L^2}$.

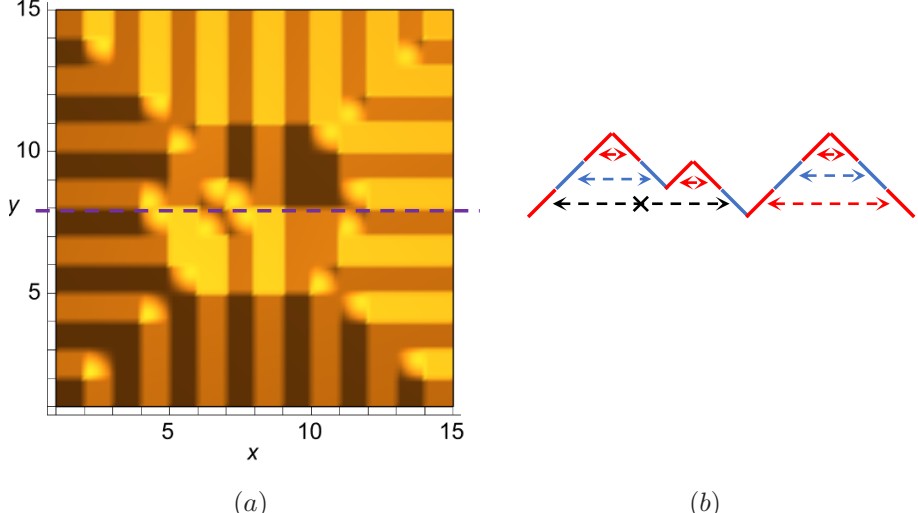

Figure 7: One particular color configuration (b) of the horizontal spins $S^{\mathrm{h}}_{x,7}$ along the cut of the dashed line in one of the 2D configuration with lowest volume (a) in the trial wave function (35) where the two endpoints of the longest nonzero height streak are mismatched in color.

However, all of them has a volume $\frac{L^3}{8}$ smaller than the maximum configuration, as depicted in Fig. 8. Lower bounding the normalization $\mathcal{N}'$ by the weight of the largest volume configuration, we have the upper bound on spectral gap

$$\langle \mathrm{ex}|\, H_{cF}\, |\mathrm{ex}\rangle < \frac{2(4\max\{(1+r^2),(1+r^{-2})\})^{L^2}}{1+q^2} q^{-\frac{L^3}{8}} \,, \tag{39}$$

which is gapless in the thermodynamic limit for the $q > 1$ phase.

## 6 Nineteen-vertex construction of coupled Motzkin chains

Building on the previous sections, we introduce a 2D generalization to the Motzkin chain, where each spin takes value of either $\pm 1$ or $0$. This can be mapped to solid edges with arrows and dashed edges without (which correspond to spin 0 in the one-dimensional chain) respectively, giving 19 vertex configurations in Fig. 9 (b) a full loop around each of which the net height change is 0, so that the height change is well-defined counting from two different paths from one plaquette to another. Nineteen-vertex model is a generalization to six-vertex model and is well-studied in the context of classical statistical mechanics [43–49]. Note that classical nineteen-vertex models are mapped to quantum spin-1 chains, by transfer matrix method, which was studied in the context of integrability [50, 51]. However, in this section, we construct a different (2D) quantum Hamiltonian that is frustration free, which enforce the ground state to be a weighted superposition between pairs of locally different configurations in Fig. 10. The boundary spins in the lattice shown in Fig. 9 (a) is enforced by boundary Hamiltonians that penalizes $-1$ spins on the left and bottom side and $+1$ on the right and top side. The bulk Hamiltonian can be defined as

$$H_{cM} = \sum_{p\in\mathrm{bulk}} \sum_{h,v=1}^{3} \sum_{c_1,c_2=\mathrm{r,b}} \frac{1}{[2]_q} \left|\pi^{c_1,c_2}_{p,h,v}\right\rangle \left\langle \pi^{c_1,c_2}_{p,h,v}\right| \,, \tag{40}$$

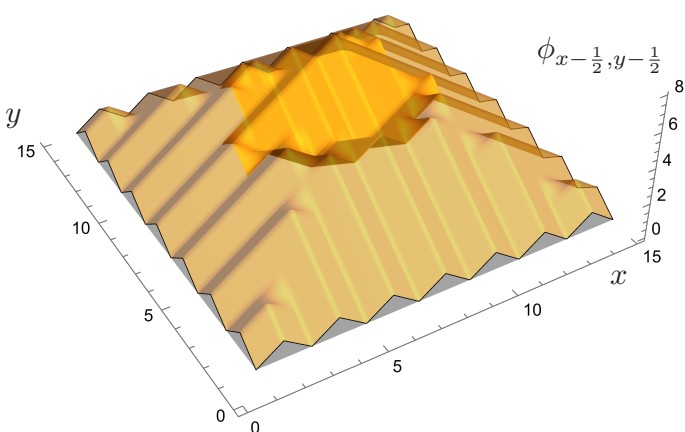

Figure 8: The volume of the dent between the minimal volume configuration in Fig. 6 and the maximal volume configuration scales as $\frac{L^3}{8}$ with the system size.

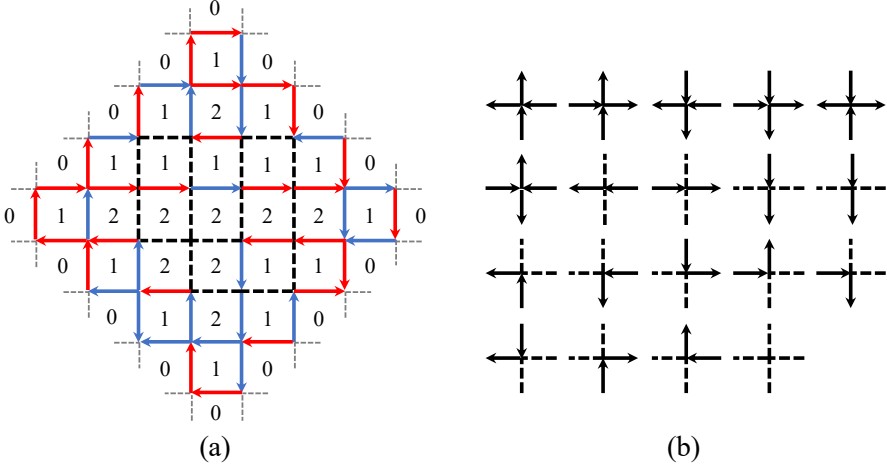

(a)                                                    (b)

Figure 9: (a) A random coupled Motzkin lattice configuration with Aztec diamond boundary. (b) The 19 vertices in the constrained Hilbert space satisfying equal number of in and out arrows.

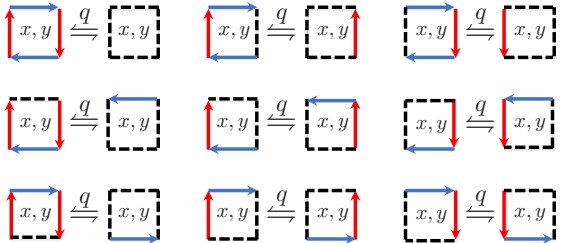

Figure 10: One coloring configuration for each of the 9 coupled Motzkin moves defined on plaquette $(x, y)$ that generate the ergodic dynamics in the Hilbert space of random surfaces of height function conditioned to be non-negative.

where

$$\left|\pi_{p,1,1}^{c_1,c_2}\right\rangle = q^{-\frac{1}{2}} \left|c_1\boxed{\begin{smallmatrix}c_2\\p\end{smallmatrix}}c_1\right\rangle - q^{\frac{1}{2}}\left|\boxed{p}\right\rangle,$$

$$\left|\pi_{p,1,2}^{c_1,c_2}\right\rangle = q^{-\frac{1}{2}} \left|c_1\boxed{\begin{smallmatrix}p\\c_2\end{smallmatrix}}c_1\right\rangle - q^{\frac{1}{2}}\left|\boxed{\begin{smallmatrix}c_2\\p\end{smallmatrix}}\right\rangle,$$

$$\left|\pi_{p,1,3}^{c_1,c_2}\right\rangle = q^{-\frac{1}{2}} \left|c_1\boxed{\begin{smallmatrix}c_2\\p\end{smallmatrix}}c_1\right\rangle - q^{\frac{1}{2}}\left|\boxed{\begin{smallmatrix}p\\c_2\end{smallmatrix}}\right\rangle,$$

$$\left|\pi_{p,2,1}^{c_1,c_2}\right\rangle = q^{-\frac{1}{2}} \left|c_2\boxed{\begin{smallmatrix}c_1\\p\end{smallmatrix}}\right\rangle - q^{\frac{1}{2}}\left|\boxed{p}\,c_2\right\rangle,$$

$$\left|\pi_{p,2,2}^{c_1,c_2}\right\rangle = q^{-\frac{1}{2}} \left|c_1\boxed{\begin{smallmatrix}p\\c_2\end{smallmatrix}}\right\rangle - q^{\frac{1}{2}}\left|\boxed{\begin{smallmatrix}c_2\\p\end{smallmatrix}}c_1\right\rangle, \qquad (41)$$

$$\left|\pi_{p,2,3}^{c_1,c_2}\right\rangle = q^{-\frac{1}{2}} \left|c_1\boxed{\begin{smallmatrix}c_2\\p\end{smallmatrix}}\right\rangle - q^{\frac{1}{2}}\left|\boxed{\begin{smallmatrix}p\\c_2\end{smallmatrix}}c_1\right\rangle,$$

$$\left|\pi_{p,3,1}^{c_1,c_2}\right\rangle = q^{-\frac{1}{2}} \left|\boxed{\begin{smallmatrix}c_1\\p\end{smallmatrix}}c_2\right\rangle - q^{\frac{1}{2}}\left|c_2\boxed{p}\right\rangle,$$

$$\left|\pi_{p,3,2}^{c_1,c_2}\right\rangle = q^{-\frac{1}{2}} \left|\boxed{\begin{smallmatrix}p\\c_2\end{smallmatrix}}c_1\right\rangle - q^{\frac{1}{2}}\left|c_1\boxed{\begin{smallmatrix}c_2\\p\end{smallmatrix}}\right\rangle,$$

$$\left|\pi_{p,3,3}^{c_1,c_2}\right\rangle = q^{-\frac{1}{2}} \left|\boxed{\begin{smallmatrix}c_2\\p\end{smallmatrix}}c_1\right\rangle - q^{\frac{1}{2}}\left|c_1\boxed{\begin{smallmatrix}p\\c_2\end{smallmatrix}}\right\rangle.$$

$H_0$ and $H_\partial$ are defined in exactly the same way as in the model of coupled Fredkin chains enforcing local constraints on the Hilbert space in Fig. 9 (b) and boundary configurations as in Fig. 9 (a), but the Hamiltonian acting on the color sector is already encoded in $H_{cM}$.

## 7 Conclusions

In this paper we have shown how quantum height models may be enhanced to give a range of exotic entropy scalings. Our models can be viewed as coupled Fredkin and Motzkin chains. They provide another example of a local Hamiltonian with volume scaling of EE. While the height degree of freedom can be described via an appropriate field theory, the addition of the color degrees of freedom within such a description is an interesting open question. Moreover the field theory description only holds for the ground state, while structure of excited states is a subject for additional work.

The equal time correlation functions of the ground state of our model are given by the correlation functions of classical six-vertex model subject to the constraint of positive height. Even in the absence of such a constraint, the analytical result of its two-point correlation functions are only computed for certain boundary conditions such as the domain wall boundary [52–54]. But it's possible to compute them numerically using Markov Chain Monte Carlo method [55]. Adding the non-negative height constraint would pose a challenge to the application of worm or loop-building algorithms, as maintaining the non-negativity would require checking a larger neighborhood as the loops get longer in each update. Another interesting next step in that direction will be the construction of a tensor network characterization for the state, as was done for in the 1D case [56, 57]. Finally, our model in the absence of an internal color degree of freedom is of interest as it promises anomalous slow dynamics and fragmentation analogous to the classical and quantum Fredkin chains in one dimension [58–60].

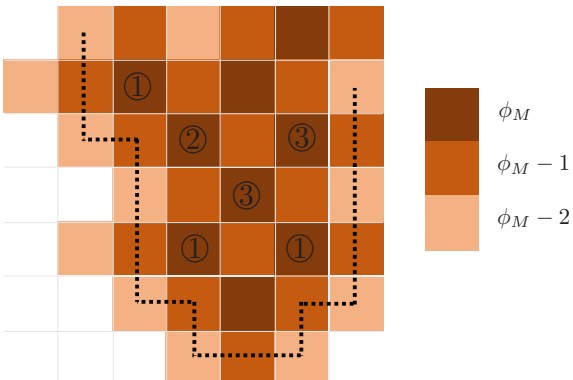

Figure 11: A snippet of plateaux contour of the maximal height $\phi_M$, with the surrounding lower height plaquettes coded by sequentially lighter shades. The heights of the plaquettes numbered ① located at the corners of the contour are ready to be lowered, while those numbered ③ are not. The plaquette ② is an accidental mobile plaquette in this contour configuration despite not lying on the corner.

## Acknowledgements

ZZ thanks Filippo Colomo, Kari Eloranta, Christophe Garban, Hosho Kastura, Yuan Miao, Henrik Røising and Benjamin Walter for fruitful discussions. ZZ acknowledges the kind hospitality of the Galileo Galilei Institute for Theoretical Physics during the workshops "Randomness, Integrability and Universality" and "Machine Learning at GGI". We gratefully acknowledge support from the Simons Center for Geometry and Physics, Stony Brook University at which some of the research for this paper was performed.

**Funding information** The work of IK was supported in part by the NSF grant DMR-1918207.

## A Ergodicity of the Hamiltonian and uniqueness of ground state

We now show that when the Hamiltonian $H_S$ acts on a properly colored height configuration it generates another such configuration, and that moreover by actions of $H_S$ we can get from any such configuration to any other. Thus the set of non-negative weighted height configurations with Dirichlet boundaries is closed under the operation, with the weighted superposition of states a unique ground state. In complete analogy with the 1D Motzkin and Fredkin chains, starting from a state which violates non-positivity in the bulk, by applying the projectors we create a superposition that will carry the negative region back to the end of the sample to get penalized by the boundary terms. Just as in the Fredkin chain case, in a non-negative height superposition involving a color violation, by reducing the height of unmatched color pairs may be pushed closer until the violation can be detected by local terms.

Let us now check that we can get to the lowest height configuration from any positive height configuration. Given any six-vertex configuration, there must be a plaquette of maximal height $\phi_M$, which may not necessarily be unique. Their nearest neighbor have height $\phi_M - 1$, but the next-nearest neighbors could either have height $\phi_M$ or $\phi_M - 2$. In the former case, we say the maximal height forms a plateau, while in the latter case, it either lies on the boundary of a plateau, or is isolated. We note that a local maximal height plaquette will have color matched pairs of edges, because of the color rule Eq. (14), therefore it can be removable by one the four moves in (16). Similarly, plaquettes that are on the the boundaries of plateaux are removable

if they are at the corner of boundaries (along a straight line of boundary, both sides in the direction of the boundary are not in the right configuration to allow one of the correlated swapping moves), since the next-nearest neighbors are both of the same height. Thus, given any boundary of a plateau, we can always reduce the volume of a surface by first removing the height cubes on the (convex) corners of plateaux boundaries, after which new corners will appear, so that the procedure keeps going. The only scenario such a procedure terminates is when the boundary forms a straight line with the plateau extending to the boundary of the lattice. In that case, both sides of the straight line have the same constant height as the boundary, meaning we have arrived at the lowest height configuration.

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
