# Peer review of "Coupled Fredkin and Motzkin chains from quantum six- and nineteen-vertex models"

_SciPost Physics, doi:SciPost Phys. 15, 044 (2023)_

## Round 3 · Referee Report · Anonymous (Referee 2) · 2023-4-2

Report

The authors have revised the manuscript and thereby improved its quality/readability substantially. In particular, the new review section about the colored-Fredkin and Motzkin chains will be beneficial to those who are unfamiliar with this subject. The authors have also added a new result concerning the spectral gap above the ground state. This is valuable, as the rigorous results for two-dimensional models are quite scarce. Besides, the authors have pertinently addressed the comments raised in my previous report. Thus, I think the current manuscript is worth publishing in SciPost. But still, I would like to ask the authors to check the following issue before publication.

- Eq. (10)
Is the Hamiltonian really correct? I know the authors corrected the sign of the fourth term in the bracket. But the subscripts of the terms ($x, y+1$, etc.) still do not match those in Fig. 1 (c). (For details, please see my comment 1 in the previous report.) The authors might want to check the consistency of this equation with the figure.

Requested changes

Minor comments:
- 1 line below Eq. (1)
Dych paths -> Dyck paths

  • validity: good
  • significance: good
  • originality: high
  • clarity: high
  • formatting: excellent
  • grammar: excellent

Author:  Zhao Zhang  on 2023-04-15  [id 3590]

(in reply to Report 1 on 2023-04-02)

We thank the referee for carefully reading the revised manuscript and for the positive report. We have noticed the mismatch between the summation range in Eq. (10) and the corresponding figure, and will correct that as well as the typo that the referee has kindly pointed out.

---

## Round 3 · Author Response

Dear editor,

Please find enclosed the revised manuscript "Coupled Fredkin and Motzkin chains from quantum six- and nineteen-vertex models" to be considered for publication in SciPost Physics. We thank the editor for the consideration of our manuscript and the referees for the helpful reports. In the revised manuscript we have considered and incorporated all points raised by the referees. With the modifications and enrichments listed below, we believe our manuscript is now ready to for the referees and the editor to make a decision.

The authors

---

## Round 3 · List of Changes

-We added a review section (Sec. 2) of the one-dimensional Fredkin and Motzkin models, to make it easier for audience unfamiliar with those models to follow the generalizations to two dimensions more easily.
-We supplemented the manuscript with a new section (Sec. 5) to discuss the scaling of the spectral gap as suggested by one of the referees.
-We reformulated the definition of the Hamiltonian terms, now Eq. (15) , (16) and (40), (41), as pointed out by one of the referees.
-We made the Schmidt decomposition, now Eq. (19) and (20), more accurate with unambiguous notation of the height function in the cross section of the bipartition.
-We gave more detail to the derivation of the decomposition of entanglement entropy in the new Eq. (27), for it to be easier to follow.
-We implemented all the other suggestions and addressed all the other concerns of the referees, a detailed list can be found in the responses to referees.

---

## Editorial Decision

published